# Mesoscopic Effects of Interfacial Thermal Conductance during Fast Pre-Melting and Melting of Metal Microparticles

Alexander Minakov [1] and Christoph Schick [2,*]

1 Prokhorov General Physics Institute of the Russian Academy of Sciences, Vavilov Str. 38, 119991 Moscow, Russia; minakov@nsc.gpi.ru
2 Institute of Physics and Competence Centre CALOR, University of Rostock, Albert-Einstein-Str. 23-24, 18051 Rostock, Germany
* Correspondence: christoph.schick@uni-rostock.de; Tel.: +49-381-498-6880

**Featured Application: The obtained knowledge can be useful for understanding and optimizing various technologies of nanostructured materials when fast melting processes take place.**

**Abstract:** Interfacial thermal conductance (ITC) affects heat transfer in many physical phenomena and is an important parameter for various technologies. The article considers the influence of various mesoscopic effects on the ITC, such as the heat transfer through the gas gap, near-field radiative heat transfer, and changes in the wetting behavior during melting. Various contributions to the ITC of the liquid-solid interfaces in the processes of fast pre-melting and melting of metal microparticles are studied. The effective distance between materials in contact is a key parameter for determining ITC. This distance changes significantly during phase transformations of materials. An unusual gradual change in ITC recently observed during pre-melting below the melting point of some metals is discussed. The pre-melting process does not occur on the surface but is a volumetric change in the microstructure of the materials. This change in the microstructure during the pre-melting determines the magnitude of the dispersion forces, the effective distance, and the near-field thermal conductance. The knowledge gained can be useful for understanding and optimizing various technological processes, such as laser additive manufacturing.

**Keywords:** interfacial thermal conductance/resistance; mesoscopic effects; near-FIELD radiative heat transfer; nanoscale heat transfer; nanostructure design; laser additive manufacturing

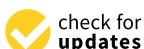





## 1. Introduction

Interfacial thermal conductance (ITC) is an important parameter for various technologies, as it affects the transfer of heat from one material to another [1]. ITC is of great importance for the development of the electronics industry, as it affects the cooling capabilities of electronic components [2]. ITC can determine the rate of fast melting and solidification of metal microdroplets during industrial laser-based processes. Thus, ITC is significant in laser welding [3] and laser additive manufacturing [4]. ITC affects the solidification rate, microstructure, and mechanical properties of the resulting materials [3,5–7]. Actually, various nanostructures can be created by controlled crystallization in many applications, such as additive manufacturing [5,8,9]. Advances in laser and additive manufacturing can be achieved through the proper development and modification of commercial base powders to optimize heat transfer in the system [10]. Thus, a deep understanding of the influence of different contributions to the ITC at the solid-solid and liquid-solid interfaces during phase transitions in microparticles can be useful for optimizing various technological processes. In this article, we will focus on the mesoscopic effects that affect the ITC during the fast

melting and solidification of metal microparticles in contact with a solid. We will consider the interfacial thermal conductance $G$, defined by Equation (1)

$$G = q/S\Delta T, \tag{1}$$

where $q$ is the heat flow across the interface, $S$ is the area of the interface, and $\Delta T$ is the temperature jump at the interface [1,11]. Heat transfer experiments in heterostructures and epitaxial interfaces (when the interfaces are atomically smooth and clean) show that ITC can be on the order of $10^8$–$10^9$ W/m$^2$K at temperatures of about 300 K [12,13]. Acoustic phonons are the dominant heat carriers through such interfaces [1,13,14]. However, most interfaces are not so perfect in the case of melting and solidification of microparticles in contact with a solid. The surface roughness of the contacting bodies leads to the appearance of an interfacial gap. The magnitude of the various contributions to ITC is highly dependent on the size of this gap. The effective distance $d$ between contacting materials is a key parameter for determining ITC. This distance can change significantly during melting and solidification. In fact, for liquid-solid interfaces, the influence of the substrate surface roughness on the ITC is strongly suppressed since liquids can spread over the surface roughness. Remarkably, liquids can spread very efficiently over surface roughness even in the absence of good wetting, as evidenced by experiments with mercury-silicon interfaces [15], as well as interfaces between water and hydrophobic surfaces [16]. Dispersion forces [17], which are always present and play an important role in interfacial adhesion during the melting of metal particles on a solid, can lead to the spreading of the sample over the surface roughness during melting. As a result, the thickness $d$ can be significantly reduced during the sample melting. Near-field photon tunneling plays an important role in ITC for liquid-solid interfaces, see below. Actually, ITC can vary over a wide range of $10^3$–$10^7$ W/m$^2$K and depends on the influence of various mesoscopic effects affecting the interfacial heat transfer.

Interfacial adhesion is the most important factor affecting ITC. For example, interfacial adhesion between nanoparticles and an organic matrix is due to the covalent chemical bonds [18,19]. This chemical adhesion is relatively strong compared to the dispersive adhesion that occurs when metal particles are melted on a solid. Chemical forces act at very short sub-nanometer distances [17]. In this case, acoustic phonons can be the dominant heat carriers at interfaces at sub-nanometer distances [14]. Interfacial phonon tunneling (with an ITC of about $10^8$ W/m$^2$K) was probably observed for aluminum and gold samples specially functionalized with organic liquids [16]. Additionally, in the case of composites of nanoparticles immersed into an organic matrix, the main contribution to the ITC, about $10^8$ W/m$^2$K, is associated with phonon heat transfer [18,19]. An interfacial thermal resistance (ITR) of about $10^{-8}$ m$^2$K/W between nanoparticles and a polymer matrix significantly affects the thermal conductivity of highly thermally conductive CNT/polymer and graphene/polymer composites [18,19]. However, in composites containing nanoparticles with lower thermal conductivity (for example, in composites and nanofluids containing aluminum oxide nanoparticles [20,21]), an ITR of about $10^{-8}$ m$^2$K/W may be insignificant for the overall thermal conductivity of the composite.

In the case of dispersive adhesion, which occurs in experiments on the melting of metal particles on a solid substrate, the ITC is not as high as in the case of chemical adhesion and is usually less than $10^7$ W/m$^2$K. In this study, we focus on the various contributions to ITC and the changes in these contributions during the pre-melting and melting of metal microparticles when dispersion adhesion takes place. In experiments with metal particles melting on a solid, we found that the main contributions to ITC are near-field photon tunneling and heat transfer through the gas (the phonon contribution to the ITC is usually not observed).

It is noteworthy that the melting of a solid is usually initiated from its surface [22], and with the onset of sample melting, a sharp jump in ITC by an order of magnitude occurs [23,24]. However, in alloys [25] and even in pure metals [26], the ITC can gradually increase upon heating in a certain temperature range below the melting point $T_{melt}$. This

recently observed unusual gradual change in ITC is analyzed in this article. It is noteworthy that this pre-melting process does not occur on the surface but is associated with the absorption of enthalpy in the volume of the sample, as follows from direct calorimetric measurements [25,26]. The purpose of this article is to study the nature of the gradual change in the ITC during the pre-melting process. In this article, it has been established for the first time that the volumetric change in the microstructure of the sample during pre-melting process determines the magnitude of the dispersion forces, the effective distance $d$, and the near-field thermal conductance.

In the first part of the article, interfacial adhesive forces and various contributions to the ITC are considered depending on the size of the interfacial gap $d$. The second part studies heat transfer across liquid-solid interfaces in melting experiments. Various mesoscopic effects affecting the ITC during phase transformations of melting microparticles are analyzed. In the last part of the article, the evolution of ITC in the processes of pre-melting and melting is studied.

## 2. Interfacial Adhesion and Dispersion Forces

Interfacial thermal conductance depends on the strength of interfacial adhesion. Adhesion forces can be divided into several types. Adhesion can be chemical, diffusive, and dispersive [17]. Chemical adhesion occurs when ionic, covalent, or hydrogen bonds are formed between the surface atoms of two bodies. Chemical forces act over very short sub-nanometer distances. Diffusive adhesion occurs when the materials can merge at an interface by diffusion. This very specific adhesion can take place when the molecules of the materials in contact are mobile and can diffuse into each other. Dispersive adhesion results from physical adsorption when two materials are attracted by long-range dispersion forces acting due to electromagnetic fluctuations [27–29]. Dispersion forces are known as electrodynamic or charge fluctuation forces. These forces contribute most to the overall van der Waals interaction of electrically neutral surfaces with nonpolar molecules. In fact, dispersion forces are the main cause of interfacial adhesion upon melting of metal particles on the surface of a solid.

We will focus on the dispersion forces $F(d)$ acting between parallel plates separated by a distance $d$. Let $p_c(d) = F(d)/S$ be the interfacial contact pressure associated with the dispersion force $F(d)$, where $S$ is the contact area. Dispersion forces are always present and play an important role both at short and long distances $d$. It is the Lifshitz theory that provides an exhaustive description of the interfacial interaction caused by dispersion forces at an arbitrary distance $d$ and for materials with arbitrary complex dielectric constants $\varepsilon_1(\omega)$ and $\varepsilon_2(\omega)$, where $\omega$ is the angular frequency of the electromagnetic wave [27–29]. Two characteristic lengths are introduced in the theory: $\lambda_T = c\hbar/k_B T$—the characteristic wavelength of thermal radiation and $\lambda_0$—the characteristic wavelength of the absorption spectra of interacting materials, where $c = 2.998 \cdot 10^8$ m/s is the speed of light in vacuum, $\hbar = 1.055 \cdot 10^{-34}$ Js is the reduced Planck constant, and $k_B = 1.38 \cdot 10^{-23}$ J/K is the Boltzmann constant [29]. It is assumed that $d$ is much larger than the interatomic distance $a$. Usually, $\lambda_T \gg \lambda_0$, where $\lambda_T$ is about 8 µm at room temperature. At large distances $d \gg \lambda_T$, the force $F(d)$ acting between two flat identical materials is about $S k_B T / 8\pi d^3$ [17,29]. However, at $\lambda_T \gg d$, the force $F(d)$ does not depend on the temperature $T$. In the case of the interaction of two identical materials with a dielectric constant $\varepsilon$, the interfacial contact pressure $p_c(d)$ can be represented by Equations (2) and (3) at $\lambda_T \gg d \gg \lambda_0$ and $\lambda_0 \gg d \gg a$, respectively. Thus,

$$p_c(d) = \frac{\pi^2 \hbar c}{240 d^4} \left( \frac{\varepsilon - 1}{\varepsilon + 1} \right)^2 f(\varepsilon), \tag{2}$$

where $f(\varepsilon)$ is of the order of 1 [17,27–29]. For example, $p_c(d) = \pi^2 \hbar c / 240 d^4$ for metal materials. This interaction is known as the Casimir force [30]. The Casimir force (per

unit area) can reach approximately $10^4$ Pa at $d = 20$ nm. In the case of $\lambda_0 \gg d \gg a$, the interfacial contact pressure $p_c(d)$ is equal to

$$p_c(d) = \frac{\hbar\omega_0}{8\pi^2 d^3}, \tag{3}$$

where $\omega_0$ is the characteristic (dominant) frequency of the absorption spectra of interacting materials [27–29]. Usually, Equation (3) is represented as

$$p_c(d) = \frac{A}{6\pi d^3}, \tag{4}$$

where the Hamaker constant $A$ is of the order of $10^{-19}$ J [17,31–33]. For example, the contact pressure $p_c(d)$ is of the order of 0.1 MPa and 5 MPa at $d = 4$ nm and 1 nm, respectively. The theory agrees well with experiments at the nanometer scale, as well as at much larger distances [31,33–35]. Thus, for mica samples, the theory gives good predictions corresponding to Equation (2) at $d > 15$ nm, and, at $d$ less than 15 nm, the interaction force is described by Equation (4) with the Hamaker constant $A = 10^{-19}$ J [31]. We will focus on dispersive adhesion associated with dispersion forces as the main cause of interfacial interaction that occurs when metal particles melt and solidify in contact with a solid.

The effective thickness $d$ of the gap between the liquid sample and the solid substrate can typically vary from a few nanometers to tens of nanometers; see below. Thus, the interfacial contact pressure $p_c(d)$ associated with dispersion forces can be in the range of $10^4$–$10^5$ Pa; see Equations (2) and (4). This pressure is sufficient to reliably attach the sample to a solid substrate and promote the spreading of the sample over the surface roughness of the substrate during melting. Thus, the thickness $d$ can be significantly reduced during the sample melting. ITC is highly dependent on the size of the interfacial gap $d$. Next, we consider the dependence of the values of various contributions to the ITC on the size of the interfacial gap $d$.

## 3. Photon and Phonon Tunneling through a Vacuum Gap

### 3.1. Nanoscale Radiative Heat Transfer

Consider heat transfer due to thermal radiation between parallel plates separated from each other by a distance $d$. In the case of a large distance $d \gg \lambda_T$, the interfacial heat flux due to radiative heat transfer in the far-field is limited by Planck's law of black-body radiation $\left(T_1^4 - T_2^4\right)\left(\pi^2 k_B^4 / 60c^2\hbar^3\right)$, where $T_1$ and $T_2$ are the temperatures of the bodies [36]. Thus, the interfacial thermal conductance associated with radiative heat transfer in the far-field does not exceed the value determined by the black-body radiation limit represented by Equation (5) [36–40]:

$$G_{bb} = \frac{\pi^2 k_B^4 T^3}{15c^2\hbar^3}, \tag{5}$$

where $G_{bb}$ is about 6 W/m$^2$K at $T = 300$ K. However, heat transfer due to thermal radiation can be increased by many orders of magnitude when the distance $d$ becomes smaller than the characteristic wavelength $\lambda_T = c\hbar/k_B T$. In this case, evanescent electromagnetic waves existing near interfaces can be transmitted between closely spaced bodies. This phenomenon, known as radiation (photon) tunneling, significantly affects the heat transfer between two bodies separated by a nanoscale gap.

### 3.2. Photon Tunneling through a Vacuum Gap

The heat transfer is greatly enhanced by the tunneling of evanescent electromagnetic waves from one body to another at $d \ll \lambda_T$ [36–41]. This is due to the fact that the number of states of evanescent electromagnetic modes localized near the surface is much higher than that of propagating waves. For example, consider electromagnetic waves near the boundary between a flat body and a vacuum. For these waves, the wave vector component $k_{||}$ parallel to the boundary must be the same on both sides of the boundary, as required by

the phase-matching boundary condition. In a vacuum, the component of the wave vector perpendicular to the surface is equal to $k_z = \sqrt{k_0^2 - k_{||}^2}$, where $k_0 = \omega/c$. For evanescent modes, the component $k_{||}$ can be much larger than the wave number of propagating modes, which is limited by $k_0$. Indeed, for evanescent modes, $k_z$ can be an imaginary number. Then, the energy density associated with evanescent modes can be much larger than that with propagating waves. However, radiation tunneling is significant only at $k_{||} \lesssim d^{-1}$. Indeed, evanescent waves in the vacuum region decay exponentially as $\exp(-z|k_z|)$ in the $z$-direction perpendicular to the surface, since $k_z$ is an imaginary number at $k_{||} > k_0$. The number of evanescent modes available to conduct heat is proportional to the volume of the phase space they occupy. This number is approximately proportional to $d^{-2}$ since $k_{||} \in (k_0, d^{-1})$, where $k_0 \ll d^{-1}$ at $d^{-1} \gg \lambda_T^{-1}$ and $\lambda_T^{-1} = k_0(k_B T/\hbar\omega)$. Thus, the effect of tunneling of evanescent waves increases approximately according to the law $1/d^2$ at $d \to 0$ [37–41]. Again, it is assumed that $d$ is much larger than the interatomic distance $a$. Additionally, in the case of metals, this increase reaches saturation when nonlocal effects become significant, that is, at $d$ around $d_F$, as determined by Equation (6) or $d \approx l_e$, the electron mean free path [36,42]:

$$d_F = \frac{v_F \hbar}{k_B T}, \tag{6}$$

where $v_F$ is the Fermi velocity [43]. For example, $d_F = 25$ nm at $v_F = 10^6$ m/s and $l_e$ is on the order of tens of nm for pure metals at room temperature, when the electron-phonon scattering predominates, and the contributions to scattering due to crystalline defects are negligible [44].

Since the density of electromagnetic energy increases significantly in the near-field of the surface, the interfacial thermal conductance $G_{nf}$ associated with near-field heat transfer is much greater than the $G_{bb}$ associated with black-body radiation. The maximum heat flux due to near-field heat transfer between two planar surfaces is approximately equal to $(k_B^2/6\hbar d^2)(T_1^2 - T_2^2)$ [37]. Thus, the maximum value of the coefficient $G_{nf}$ can be represented by Equation (7).

$$G_{nf}^{max}(d) = \frac{k_B^2 T}{\hbar d^2} \mathrm{B}_{max}, \tag{7}$$

where $\mathrm{B}_{max} = 1/3$. For example, $G_{nf}^{max}(d)$ is about $10^6$ W/m$^2$K at $d = 10$ nm and $T = 300$ K. However, not every evanescent mode available for thermal conduction effectively participates in the heat transfer. Actually, $G_{nf}(d)$ is approximately one or two orders of magnitude less than $G_{nf}^{max}(d)$ for insulators [37–39,45–47] or semiconductors [48–52] and about two or three orders of magnitude less than $G_{nf}^{max}(d)$ for metals [37,42,45,53–55]. However, the near-field heat transfer at the metal-dielectric interface is greater than at the metal-metal interface, and this heat transfer can be further enhanced by an oxide film on the metal surface. This enhancement is significant if the thickness of the oxide films exceeds the size of the interfacial gap [45].

The efficiency of heat transfer by evanescent modes depends on the dielectric properties of the interacting materials. For example, resonant surface waves (surface phonon polaritons) make a significant contribution to heat transfer between polar dielectric materials with complex dielectric constants $\varepsilon_1(\omega)$ and $\varepsilon_2(\omega)$ at $d \ll \lambda_T$. In this case, the spectral density $G_{nf}(\omega)$ of the interfacial thermal conductance $G_{nf}$ can be approximately represented by Equation (8) [38–41].

$$G_{nf}(\omega) \simeq \frac{k_B}{d^2} \frac{\mathrm{Im}(\varepsilon_1)\,\mathrm{Im}(\varepsilon_2)}{|1 + \varepsilon_1|^2 |1 + \varepsilon_2|^2} \left(\frac{\hbar\omega}{k_B T}\right)^2 \frac{\exp\left(\frac{\hbar\omega}{k_B T}\right)}{\left[\exp\left(\frac{\hbar\omega}{k_B T}\right) - 1\right]^2}, \tag{8}$$

Denote $B = \int \frac{\text{Im}(\varepsilon_1)\,\text{Im}(\varepsilon_2)}{|1+\varepsilon_1|^2\,|1+\varepsilon_2|^2} \frac{x^2 e^x}{(e^x-1)^2} dx$, where $x = \hbar\omega/k_B T$, then

$$G_{nf}(d) \simeq \frac{k_B^2 T}{\hbar d^2} B, \qquad (9)$$

where the coefficient B depends on the complex dielectric constants $\varepsilon_1(\omega)$ and $\varepsilon_2(\omega)$. Thus, heat transfer by photon tunneling is very efficient at frequencies close to surface polariton resonances, at which $\text{Re}\varepsilon_1(\omega)$ or $\text{Re}\varepsilon_2(\omega)$ approaches $-1$; see Equation (8). Surface wave resonances are the main channels of heat transfer in the near-field. In fact, the energy transfer mediated by evanescent tunneling modes was significantly enhanced due to surface phonon polaritons in dielectrics and semiconductors [37–40,45–47], surface plasmon polaritons (which can be observed at interfaces containing metals or doped semiconductors) [37,45–56], and vibrational states of adsorbates on the surface of various types of materials [57]. The near-field photon tunneling is significant at distances $d$ below 100 nm. However, at sub-nanometer distances, low-frequency acoustic phonons can tunnel though the vacuum gap by coupling with evanescent electric fields.

### 3.3. Phonon Coupling at Sub-Nanometer Gaps

As the interfacial gap decreases, the near-field heat transfer is replaced by the tunneling of acoustic phonons through sub-nanometer gaps. Phonon tunneling can be explained by the modulation of interfacial dispersion forces due to surface vibrations. Thus, lattice vibrations can be transmitted through a very thin vacuum gap between contacting bodies. It is noteworthy that phonon and photon tunneling can be described by a single formalism, which demonstrates the complete picture of the transition from heat transfer by evanescent thermal radiation to phonon heat conduction [14]. Low-frequency acoustic phonons tunnel through the vacuum gap due to interaction with damped electric fields. Thus, an additional heat transfer channel is formed. The phonon tunneling provides a significant increase in heat transfer compared to photon tunneling at sub-nanometer gaps. Thus, there is a transition from near-field thermal radiation to phonon heat conduction, and acoustic phonons become the dominant heat carriers at $d < 1$ nm [14]. This effect can lead to an increase in the interfacial thermal conductance to a value associated with phonon tunneling $G_{ph}$, which is about $10^7$–$10^8$ W/m$^2$K at $d < 1$ nm and $T > 100$ K [14].

## 4. Heat Transfer through the Gas Gap

In experiments on melting and solidification, the interfacial gas gap depends on the state of the melting substance and the roughness of the contacting bodies. The effective thickness $d$ of this gap can be in the range of 1–100 nm for liquid-solid interfaces. Note that the mean free pass $l_{mfp}$ of nitrogen gas molecules is on the order of 100 nm at $T > 300$ K [58]. Thus, in order to estimate the interfacial thermal conductance $G_g$ associated with heat transfer through the gas, we consider the case of heat exchange between two plates separated by a thin gap of thickness $d \ll l_{mfp}$. Since the experiments considered below were conducted in air at elevated temperatures, we make estimates at $T > 300$ K for nitrogen gas, which is the main component of air.

Consider two plates heated to different temperatures $T_1$ and $T_2$, with $(T_1 - T_2) \ll T$, where $T$ is the intermediate of $T_1$ and $T_2$. Note that the heat transfer at the boundary due to natural convection in gases is negligibly small compared to the thermal conductivity of gases at a microscale distance [59]. The convective component of heat transfer in gases arises in a gravitational field in the presence of temperature gradients due to the temperature dependence of the gas density. The convective component decreases with the decreasing characteristic length $L$ of the problem. The ratio of the convective to conductive components of heat transfer is equal to the Nusselt number $\text{Nu} = \left(\text{Gr}\right)^{1/4} f(\text{Pr})$ [60]. The Prandtl number Pr and the value of the function $f(\text{Pr})$ for gases are about 1 [58,60]. The Grashof number for gases is approximately equal to $\text{Gr} = L^3 g (T_1 - T_2)/\nu^2 T$, where $g$ is the gravitational acceleration, and $\nu$ is the kinematic viscosity of the gas [60]. Thus, the

Nusselt number decreases proportionally to $L^{3/4}$ as $L$ decreases. Nu is about $0.8 \cdot 10^{-2}$ at $L = 1 \, \mu m$, $(T_1 - T_2)/T = 0.1$, and $\nu = 1.6 \cdot 10^{-5} \, m^2/s$ for nitrogen gas at normal pressure and room temperature [58]. Therefore, the convective contribution to heat transfer is less than 1% at $d < 1 \, \mu m$ (this contribution is about 4% at $L$ about $10 \, \mu m$).

Let us first assume that the gas molecules interact with the plates under conditions of complete accommodation, i.e., the molecules reflected from a body have the same temperature as the surface of this body. The amount of heat transferred by the gas molecules between the plates is proportional to the number of molecules striking a unit area of the surface per unit time $Z = nV_m/4$, where $n = p/k_B T$ is the number of molecules per unit volume at pressure $p$ and $V_m = \sqrt{8k_B T/\pi m}$ is the average velocity of gas molecules of a mass $m$. Thus, $Z = p/\sqrt{2\pi m k_B T}$ [61]. The heat flux carried away by gas molecules from the surface is equal to $q_g = Z(c_v + k_B/2)T$, where $c_v$ is the heat capacity per one molecule at a constant volume. This result is based on the assumption of a Maxwellian velocity distribution [61]. Then, the thermal conductance $G_g = \Delta q_g/(T_1 - T_2)$ between plates 1 and 2 due to the gas can be represented by Equation (10).

$$G_g = (c_v + \frac{k_B}{2}) \frac{p}{\sqrt{2\pi m k_B T}}. \tag{10}$$

Thus, $G_g$ does not depend on $d$ since $d$ is much smaller than the transverse dimension of the plates and $d \ll l_{mfp}$.

Consider the case of incomplete accommodation of gas molecules with the surfaces of the plates. Then, there are discontinuities of the temperature at the surfaces. The molecules rarely interact with each other but interact mainly with the surfaces of the plates at $d \ll l_{mfp}$. Thus, it is necessary to clarify the concept of gas temperature $T_g$. Usually, $T_g$ is defined as the average energy $E_g$ of molecules at a given point in the gas, and the relationship between the gas temperature and energy is the same as for large gas volumes [61,62]. Denote by $E_{g1}$ and $E_{g2}$ the energies of gas flows moving from plate 1 to plate 2 and back, respectively. $T_{g1}$ and $T_{g2}$ are the temperatures of these flows. Let $E_1$ and $E_2$ be the energies of similar gas flows with temperatures $T_1$ and $T_2$. The heat flux $q_g$ carried by the flows between the plates is proportional to the energy difference $E_{g1} - E_{g2}$. The discontinuities of the temperature at the surfaces of the plates are proportional to this heat flux. Let $T_1 > T_2$. Then, the boundary conditions at the plates can be represented by Equations (11) [62].

$$\sigma_1(E_1 - E_{g2}) = (E_{g1} - E_{g2}), \tag{11a}$$

$$\sigma_2(E_{g1} - E_2) = (E_{g1} - E_{g2}), \tag{11b}$$

where $\sigma_1$ and $\sigma_2$ are the thermal accommodation coefficients of gas molecules on the surfaces of plates 1 and 2. Therefore, $(E_{g1} - E_{g2}) = \frac{\sigma_1\sigma_2}{(\sigma_1+\sigma_2-\sigma_1\sigma_2)}(E_1 - E_2)$, as follows from the boundary conditions (11). Usually, the coefficients $\sigma_1$ and $\sigma_2$ are difficult to determine separately for thermal contact surfaces. However, the thermal contact can be characterized by the combined coefficient $\sigma = \frac{2\sigma_1\sigma_2}{(\sigma_1+\sigma_2)}$. Then, we obtain $\frac{\sigma_1\sigma_2}{(\sigma_1+\sigma_2-\sigma_1\sigma_2)} = \frac{\sigma}{(2-\sigma)}$. The thermal conductance $G_g = \Delta q_g/(T_1 - T_2)$ between plates 1 and 2 can be obtained from Equation (10) by multiplying by the coefficient $\frac{\sigma}{(2-\sigma)}$; see Equation (12).

$$G_g = (c_v + \frac{k_B}{2}) \frac{\sigma}{(2-\sigma)} \frac{p}{\sqrt{2\pi m k_B T}}. \tag{12}$$

The accommodation coefficient $\sigma$ is about 0.6 for nitrogen gas in contact with a clean surface [11,63]. Thus, $G_g$ is about $5.1 \cdot 10^4 \, W/m^2 K$ at $p = 10^5$ Pa and $T = 300$ K for nitrogen gas with $\sigma = 0.6$, $c_v = 5k_B/2$, and $m = 4.65 \cdot 10^{-26}$ kg [58].

In the case of a thick gas gap $d$, the coefficient $G_g$ can be represented by Equation (13)

$$G_g(d) = \frac{\varkappa_g}{d + \delta_1 + \delta_2}, \tag{13}$$

where $\varkappa_g$ is the gas thermal conductivity [11]. The distances $\delta_i$ associated with the temperature jumps on the first or second surfaces are represented by Equation (14) [11].

$$\delta_i = \frac{(2 - \sigma_i)}{\sigma_i} \frac{2\gamma}{(1 + \gamma)} \frac{l_{mfp}}{\text{Pr}}, \tag{14}$$

where $\gamma = C_p / C_v$ is the ratio of the specific heats at constant pressure and constant volume. Considering that $l_{mfp} = \nu\sqrt{\pi m / 2k_B T}$, $\text{Pr} = \nu\rho C_p / \varkappa_g$, and $\rho$ is the gas density [61,62], we obtain Equation (15).

$$\delta_i = \frac{(2 - \sigma_i)}{\sigma_i} \frac{\varkappa_g}{2c_v + k_B} \frac{\sqrt{2\pi m k_B T}}{p}, \tag{15}$$

Thus, as follows from Equation (15), Equation (13) transforms into Equation (12) for $d \to 0$, and $\sigma_1 = \sigma_2 = \sigma$. $G_g(d)$ increases with decreasing gas gap distance $d$ according to Equation (13) and reaches the maximum possible value $G_g^{max}$ at $d \to 0$, where $G_g^{max}$ is determined by Equation (12).

This theory is well demonstrated in the following experiment, in which the ITC between two flat specimens of C45 steel was measured in real time while cycling the contact pressure $p_c$ at $T$ about 350 K in air at atmospheric pressure [64]. An increase in ITC from $0.8 \cdot 10^4$ to $3.2 \cdot 10^4$ W/m$^2$K was observed when the contact pressure $p_c$ changed from 25.8 to 120 MPa. Then, measurements were conducted with a cyclic change in the contact pressure $p_c$ in the range of 50–120 MPa. The ITC varied within $3 \cdot 10^4$–$3.2 \cdot 10^4$ W/m$^2$K with such a cyclic change in pressure. The initial surface roughness of the samples was $R_a = 6.5$ μm. However, during loading cycles, both surfaces of the samples were flattened due to elastoplastic deformations. This led to a decrease in the effective distance $d$ and an increase of ITC. The measurements agree well with Equations (13) and (15) for a reasonable effective thickness $d$. Indeed, $G_g(d) = 0.8 \cdot 10^4$ W/m$^2$K and $3.2 \cdot 10^4$ W/m$^2$K at $d = 3$ μm and 0.3 μm for the beginning of the first cycle and after the loading cycles, respectively. Calculations according to Equation (13) were carried out for nitrogen gas at $\sigma = 0.6$, $p = 10^5$ Pa, $T = 350$ K, $\varkappa_g(T) = 0.029$ W/m · K [58], and $\delta = 0.3$ μm, see Equation (15). The data are collected in Table 1.

**Table 1.** Thermal contact parameters of steel specimens with initial surface roughness $Ra = 6.5$ μm.

| Contact Pressure $p_c$ MPa | ITC $G_g(d)$ W/m$^2$K | Effective Thickness $d$ μm | Temperature Jump Distance $\delta$ μm | Temperature $T$ K | Gas Pressure $p$ Pa | Gas Thermal Conductivity $\varkappa_g(T)$ W/m·K |
|---|---|---|---|---|---|---|
| 25.8 | $0.8 \cdot 10^4$ | 3.0 | 0.3 | 350 | $10^5$ | 0.029 |
| 120 | $3.2 \cdot 10^4$ | 0.3 | 0.3 | 350 | $10^5$ | 0.029 |

The interfacial thermal conductance through the gas $G_g(d)$ provides the main contribution to the total interfacial thermal conductance at $d > 100$ nm; see Figure 1. However, at $d$ of the order of 100 nm, there is a transition from heat conduction through gas to heat conduction by the near-field thermal radiation. Thus, at distances $d$ less than 100 nm, near-field photon tunneling becomes the most significant channel of interfacial heat transfer; see Figure 1.

In experiments on melting and solidification, the thickness $d$ depends on the state of the melting sample and the roughness of the solid substrate. The thickness $d$ can be much less than the substrate roughness since the liquid can spread over surface asperities. Thus, the interfacial thermal conductance $G_{LS}$ of the liquid-solid interfaces is generally much greater than the interfacial thermal conductance $G_{SS}$ of the solid-solid interfaces. In fact, the interfacial gap decreases significantly during melting, and $G_{LS}$ is usually due to the near-field thermal conductance $G_{nf}$, while $G_{SS}$ is due to the thermal conductance through the gas $G_g$.

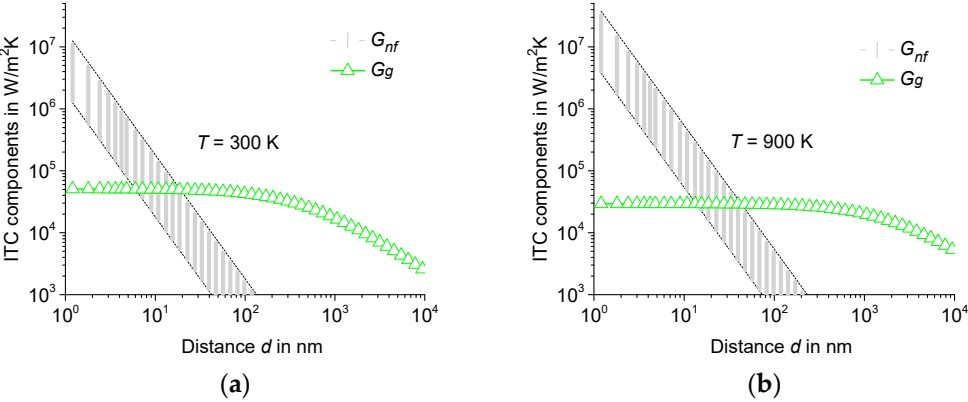

**Figure 1.** ITC components $G_{nf}(d)$ and $G_g(d)$ vs. distance $d$ for flat surfaces in nitrogen gas at $\sigma = 0.6$, $p = 10^5$ Pa, and $T = 300$ K (**a**), as well as $T = 900$ K (**b**). The near-field component is estimated as $G_{nf} \in \left( G_{nf}^{max}/100, \, G_{nf}^{max}/10 \right)$.

## 5. Heat Transfer across Liquid-Solid Interfaces

### 5.1. Effective Gap Thickness at the Liquid-Solid Interface

The interface between the contacting materials can change significantly during melting and solidification. If the surface roughness approaches the submicron scale during melting, near-field thermal radiation can play a significant role in the overall heat transfer. Consider a thermal contact between a molten sample and a solid substrate. It can be expected that for the liquid-solid interfaces, the effect of substrate surface roughness on ITC will be strongly suppressed. Indeed, liquid samples can spread well over surface irregularities. Moreover, liquids can spread very effectively over surface roughness, even in the absence of good wetting. For example, at the mercury-silicon interface, $G_{LS}$ is about $10^6$ W/m$^2$K for mercury microdroplets at $T = 450$ K and a low gas pressure $p$ of about 60 Pa [15]. In this case, the interfacial thermal conductance through the gas is negligible. Actually, $G_g^{max}$ determined by Equation (12) is about 25 W/m$^2$K for nitrogen gas at $\sigma = 0.6$, $p = 60$ Pa, and $T = 450$ K. However, the interfacial thermal conductance of about $10^6$ W/m$^2$K can be achieved due to the near-field thermal conductance $G_{nf}(d)$. The near-field thermal conductance $G_{nf}(d)$ can be approximately estimated between the values $G_{nf}^{max}(d)/100$ and $G_{nf}^{max}(d)/10$. Then, from Equation (7), we obtain that $G_{nf}(d)$ is about $10^6$ W/m$^2$K at $d$ in the range of 1–5 nm. Thus, the effective gap $d$ at the interface between mercury and silicon substrate can be estimated to be a few nanometers. Data on the interfacial thermal contact of mercury and silicon are collected in Table 2.

**Table 2.** Thermal contact parameters of the mercury-silicon interface at low gas pressure.

| ITC $G_{LS}(d)$ W/m$^2$K | ITC Component $G_g^{max}$ W/m$^2$K | ITC Component $G_{nf}(d)$ W/m$^2$K | Effective Thickness $d$ nm | Temperature $T$ K | Gas Pressure $p$ Pa |
|---|---|---|---|---|---|
| $10^6$ | 25 | $10^6$ | 1–5 | 450 | 60 |

In the case of liquid–solid interfaces, the effective thickness $d$ is much less than the substrate surface roughness $R_a$, as follows from $G_{LS}$ measurements for molten copper droplets cooling on cold copper substrates [23]. Furthermore, the effect of surface roughness on the ITC of liquid-solid interfaces is rather weak. Indeed, $G_{LS}$ increases only by a factor of 2 (from $4.8 \cdot 10^4$ to $9 \cdot 10^4$ W/m$^2$K) with a decrease in surface roughness $R_a$ by two orders of magnitude (from 7.7 to 0.07 μm) [23]. In this experiment, the gas component $G_g(d)$ can contribute to the total ITC no more than the maximum possible value $G_g^{max} = 2.5 \cdot 10^4$ W/m$^2$K at $d \to 0$, $\sigma = 0.6$, $p = 10^5$ Pa, and $T = 1500$ K. In fact, $G_g(d)$ is approximately equal to $G_g^{max}$ at $d < 100$ nm since the dependence $G_g(d)$ saturates at $d$ less than 100 nm; for ex-

ample, see Figure 1. The near-field component $G_{nf}(d)$ can be approximately estimated between the values $G_{nf}^{max}(d)/100$ and $G_{nf}^{max}(d)/10$. Thus, $G_{nf}(d) = G_{LS} - G_g^{max}$ is about $2.3 \cdot 10^4$ W/m²K at $d = 20$–62 nm for substrates with surface roughness $R_a = 7.7$ μm. Additionally, $G_{nf}(d) = G_{LS} - G_g^{max}$ is about $6.5 \cdot 10^4$ W/m²K at $d = 12$–37 nm for substrates with $R_a = 0.07$ μm. Data on ITC parameters for copper-copper interfaces at different surface roughness are collected in Table 3.

**Table 3.** ITC parameters of Cu/Cu interfaces at different surface roughness, $T$ about 1500 K, and $10^5$ Pa.

| Surface Roughness $Ra$ μm | ITC $G_{LS}(d)$ W/m²K | ITC Component $G_g^{max}$ W/m²K | ITC Component $G_{nf}(d)$ W/m²K | Effective Thickness $d$ nm |
|---|---|---|---|---|
| 7.7 | $4.8 \cdot 10^4$ | $2.5 \cdot 10^4$ | $2.3 \cdot 10^4$ | 20–62 |
| 0.07 | $9 \cdot 10^4$ | $2.5 \cdot 10^4$ | $6.5 \cdot 10^4$ | 12–37 |

Similar results were obtained when measuring $G_{LS}$ for molten nickel cooled on cold substrates of copper, aluminum, and stainless steel with different roughness [23]. For example, $G_{LS}$ increases from $2 \cdot 10^4$ W/m²K to $3.5 \cdot 10^4$ W/m²K with a decrease in $R_a$ from 1.2 μm to 0.18 μm for aluminum substrates, from $3.5 \cdot 10^4$ W/m²K to $1.2 \cdot 10^5$ W/m²K with a decrease in $R_a$ from 1.5 μm to 0.17 μm for steel substrates, and from $4 \cdot 10^4$ W/m²K to $2.6 \cdot 10^5$ W/m²K with a decrease in $R_a$ from 7.7 μm to 0.07 μm for copper substrates [23].

Thus, in the case of the liquid-solid interfaces, the effective thickness $d$ is much less than the surface roughness $R_a$; for example, see Table 3. In fact, liquids spread very efficiently over surface roughness even in the absence of good wetting [15,16]. This interesting fact explains the significant difference between the surface roughness $R_a$ and the effective distance $d$ in experiments on the melting of various materials. However, in melting experiments, the distance $d$ cannot tend to zero since the surface roughness usually has a multi-scale structure and liquids cannot spread over too small surface irregularities.

The near-field thermal conductance $G_{nf}(d)$ at nanometer distances $d$ significantly exceeds the maximum possible value $G_g^{max}$ associated with thermal conduction through the gas, see Figure 1. Moreover, as the distance $d$ approaches the sub-nanometer scale, the ITC can be significantly increased due to the tunneling of low-frequency acoustic phonons. Indeed, $G_{ph}(d)$ is about $10^7$–$10^8$ W/m²K at $d < 1$ nm [14]. However, phonon tunneling is usually not observed in melt-solidification experiments. In fact, liquids cannot spread over too-fine surface irregularities less than 1 nm in size. Thus, in melt-solidification experiments, the distance $d$ is usually at least 1 nm or more. The main contributions to the ITC at melting experiments is made by the following two components: $G_{nf}(d)$ and $G_g(d)$, associated with the near-field thermal conductance and thermal conductance through the gas, respectively; see Figure 1.

However, phonon tunneling was probably observed in $G_{LS}$ measurements for water in contact with specially functionalized aluminum and gold substrates. Remarkably, for both hydrophilic and hydrophobic substrates, $G_{LS}$ is of the order of $10^8$ W/m²K [16]. Indeed, $G_{LS} = 10^8$ W/m²K and $1.8 \cdot 10^8$ W/m²K for hydrophilic surfaces of gold and aluminum functionalized with 11-mercapto-1-undecanol and 2-[methoxy(polyethyleneoxy)-propyl]-trichlorosilane, respectively. The thickness of the functional layers was 1.1 nm and 0.7 nm for gold and aluminum, respectively. In the case of hydrophobic surfaces, $G_{LS}$ is equal to $5 \cdot 10^7$ W/m²K and $6 \cdot 10^7$ W/m²K for gold and aluminum functionalized with 1-octadecanethiol and octadecyl trichlorosilane, respectively. In both cases, the thickness of the functional layers was about 2.3–2.4 nm [16]. $G_g^{max}$ determined by Equation (12) is about $5.1 \cdot 10^4$ W/m²K for nitrogen gas at $d \to 0$, $\sigma = 0.6$, $p = 10^5$ Pa, and $T = 300$ K, and $m = 4.65 \cdot 10^{-26}$ kg [58]. For saturated water vapor at 30C, the maximum possible value $G_g^{max} = 7.2 \cdot 10^3$ W/m²K at $d \to 0$, $\sigma = 1$, $c_v = 3k_B$, $p = 4.2 \cdot 10^3$ Pa, $T = 303$ K, and

$m = 2.99 \cdot 10^{-26}$ kg [58]. Thus, the heat conduction through the gas is negligible. The value of the near-field component $G_{nf}(d)$ can be of the order of $10^8$ W/m$^2$K at $d = 0.1$–0.4 nm. However, phonon tunneling $G_{pn}$ can also contribute about $10^8$ W/m$^2$K to ITC at such a short distance [14].

Recent measurements of the time dependences of ITC with sub-millisecond resolution made it possible to separate the processes before the onset of melting (during pre-melting) and during the melting of metal microparticles, as well as to study the evolution of ITC during pre-melting [24–26]. A gradual change in ITC is observed below the melting point of some metals. In the following, we will focus on studying the various contributions to ITC during the melting and pre-melting processes.

### 5.2. Evolution of Thermal Contact Conductance G during the Pre-Melting Process

Generally, ITC at the interface between the sample and the substrate abruptly jumps by an order of magnitude when the measured sample melts or solidifies, as was observed in experiments with copper, nickel, and tin [23,24]. Usually, the melting of a solid starts from its surface [22]. Therefore, the ITC jump from $G_{SS}$ to $G_{LS}$ occurs just at the beginning of the melting process due to the formation of a thin liquid layer on the sample surface. Thus, direct calorimetric measurements of the enthalpy absorbed by a sample of pure tin in melting experiments definitely indicate that the melting process begins with a very thin surface layer and not in the bulk of the sample [24]. However, in the case of pure indium, as well as an aluminum alloy (AA7075), ITC gradually increases upon heating in a certain temperature range below the melting point $T_{melt}$. Moreover, this pre-melting process occurs in the bulk of the sample as follows from direct calorimetric measurements [25,26]. Next, we consider the various contributions to the interfacial thermal conductance with a gradual change in ITC during the pre-melting process. Ultrafast nanocalorimetry combined with high-speed IR thermography was used to measure the ITC of microparticles of several nanograms in fast thermal processes at a temperature scan rate in the range of $10^2$–$10^5$ K/s [25].

The calorimetric sensor consists of an amorphous Si-N membrane (about 1 μm thick) with a resistive heater and a thermocouple sensor located in the central part of the membrane directly below the sample. The temperature sensor, heater, and electrical connections are formed by thin-film tracks of doped p-type and n-type polysilicon [25,26]. Ultrafast scanning nanocalorimetry makes it possible to measure the dynamics of the enthalpy absorbed by a sample during fast phase transformations with sub-nanojoule resolution. Measurements can be made at temperature scan rates up to $10^8$ K/s [65]. The dynamics of the temperature on the sample surface opposite the membrane was measured by IR thermography with sub-millisecond time resolution [25,26]. Thus, the temperature difference between the sample and the membrane was measured and used to determine the ITC, since the temperature difference across the sample (less than 1 K) was small compared to the measured temperature difference across the membrane-sample interface. The measurements were completely reproducible during successive heating-cooling temperature scans. The calorimetric measurement of the heat flux through the membrane-sample interface was conducted with an error of about 10%. The ITC measurement error was about 50% or less, depending on the scan rate and temperature range. This error was mainly associated with the uncertainty of the interfacial temperature difference of about 30%. Thus, according to Equation (1), the ITC was measured during fast phase transformations in metal microdroplets with a sub-millisecond resolution [25,26].

The uncertainty in the distance $d$ estimated from the measured ITC is defined as follows. At distances $d > 100$ nm, the ITC is less than $G_g^{max}$, and the ITC is determined mainly by the component $G_g(d)$; see Figure 1. In this case, the distance $d$ is calculated using Equations (13) and (15) with an error of about 45%. This error is mainly due to the uncertainty of the accommodation coefficient $\sigma$, which is about 30%. The error of gas thermal conductivity $\varkappa_g$ (about 3%) is insignificant. This error is associated with the replacement of the thermal conductivity of air with the thermal conductivity of nitrogen gas [58]. The overall error in estimating $d$ is about 70% because ITC is measured with an uncertainty of

about 50%. At distances $d$ less than 100 nm, the interfacial thermal conductance $G_{LS}$ can be represented as the sum of $G_g(d)$ and $G_{nf}(d)$, where the near-field component $G_{nf}(d)$ is estimated as $G_{nf} \in \left( G_{nf}^{max}/100, G_{nf}^{max}/10 \right)$, and $G_g(d)$ is approximately equal to $G_g^{max}$, see Figure 1. In this case, $G_{nf}$ is estimated with an uncertainty of about 50%. The uncertainty of $G_g^{max}$ is about 45%, mainly due to the uncertainty of $\sigma$ of about 30%. The total error in estimating $d$ is about 80% because ITC is measured with an uncertainty of about 50%. However, the accuracy of the estimate $d$ is quite satisfactory since the distance $d$ changes by orders of magnitude in melting experiments (note the logarithmic scales in Figure 1).

Measurements on a millisecond time scale made it possible to separate the influence of the pre-melting and melting processes on the ITC. These measurements are conducted in calorimetric microchips, in which the sample is placed on an amorphous silicon nitride membrane with a surface roughness of about 200 nm [66]. This roughness, if necessary, can be reduced by chemical etching to about 1–10 nm [67,68]. The roughness of the sample may be greater than that of the membrane at the beginning of sample melting. However, as the sample melts, this roughness decreases, and, finally, the interfacial gap thickness $d$ can be much smaller than the membrane roughness.

Consider experiments on the melting of microsamples of pure indium in contact with the membrane of a calorimetric sensor [26]. Indium microsamples were heated from 320 K to 550 K (above the melting point $T_{melt} = 430$ K) at a rate of about $2 \cdot 10^3$ K/s and cooled back to 320 K at the same rate. In these experiments, the position and shape of the sample remained stable after the first melt, and the measurements were completely reproducible during subsequent heating-cooling scans. It was found that the melting process is divided into two stages: the pre-melting process and the actual melting process. At the beginning of the pre-melting process (at $T_{melt} - 20$ K), the ITC sharply increases from $G_{SS} = 2.7 \cdot 10^3$ W/m$^2$K to $G_{pS} = 4.1 \cdot 10^3$ W/m$^2$K. This jump in ITC is associated with a change in the surface of the sample, as indicated by calorimetric measurements [26]. Further, within 2.6 ms in the temperature range $(T_{melt} - 20$ K, $T_{melt})$, a volumetric pre-melting process occurs with an enthalpy consumption, which is clearly seen from measurements of the enthalpy absorbed by the pre-melting sample [26]. ITC gradually increases from $G_{pS} = 4.1 \cdot 10^3$ W/m$^2$K to $G_{LS} = 5 \cdot 10^4$ W/m$^2$K during this pre-melting process. After that, the actual melting occurs for 1.6 ms at $T_{melt} = 430$ K. The ITC remains constant $(G_{LS} = 5 \cdot 10^4$ W/m$^2$K) during this melting process [26].

In this experiment, $G_{SS} = 2.7 \cdot 10^3$ W/m$^2$K corresponds to the ITC for the solid-solid interface between the indium sample and the calorimeter membrane. This value $(2.7 \cdot 10^3$ W/m$^2$K) can be attributed to the heat conduction through the nitrogen gas at $d = 11.5$ μm, $\sigma = 0.6$, $p = 10^5$ Pa, $T_{int} = 420$ K, $\delta = 0.39$ μm, and $\varkappa_g(T_{int}) = 0.034$ W/m $\cdot$ K [58]; see Equations (13) and (15), where $T_{int}$ is the temperature intermediate between the sample temperature $T_S = 410$ K and the membrane temperature $T_m = 430$ K. Similarly, $G_{pS}(d) = 4.1 \cdot 10^3$ W/m$^2$K can be attributed to the heat conduction through the nitrogen gas at $d = 7.4$ μm and $T_{int} = 420$ K. The near-field component is negligible at $d = 7.4$ μm.

However, with subsequent heating, the ITC gradually increases and reaches the value $G_{LS} = 5 \cdot 10^4$ W/m$^2$K at the melting point. This value exceeds the ITC associated with the gas thermal conduction at $T_{int}$ about 433 K, where $T_{int}$ is the temperature intermediate between $T_S = 430$ K and $T_m = 435$ K. The interfacial thermal conductance $G_{LS} = 5 \cdot 10^4$ W/m$^2$K can be represented as the sum of $G_g(d)$ and $G_{nf}(d)$, where $G_{nf}(d)$ is about $10^4$ W/m$^2$K at $d = 17$–50 nm, and $G_g(d)$ is about $4 \cdot 10^4$ W/m$^2$K at $d \leq 50$ nm, $\sigma = 0.6$, $p = 10^5$ Pa, $T_{int} = 433$ K, $\delta = 0.41$ μm, and $\varkappa_g(T_{int}) = 0.035$ W/m·K [58]; see Equations (13) and (15). ITC parameters at the indium-membrane interface for the beginning and end of the pre-melting temperature range are collected in Table 4.

**Table 4.** ITC parameters at the indium-membrane interface in the pre-melting temperature range near $T_{melt} = 430$ K.

| Sample Temperature $T_S$ K | Intermediate Temperature $T_{int}$ K | ITC $G_{LS}(d)$ W/m$^2$K | ITC Component $G_g(d)$ W/m$^2$K | ITC Component $G_{nf}(d)$ W/m$^2$K | Effective Thickness $d$ nm |
|---|---|---|---|---|---|
| $T_{melt} - 20$ K | 420 | $4.1 \cdot 10^3$ | $4.1 \cdot 10^3$ | <0.5 | 7400 |
| $T_{melt}$ | 433 | $5 \cdot 10^4$ | $4 \cdot 10^4$ | $1 \cdot 10^4$ | 17–50 |

Thus, in the process of pre-melting of pure indium, the distance $d$ decreases from about ten micrometers to tens of nanometers, while the ITC increases by an order of magnitude. Note that at the beginning of the pre-melting process, the distance $d$ significantly exceeds the membrane surface roughness $R_a = 0.2$ μm. Therefore, the surface of the sample may be in a pasty state rather than in a liquid state. The sample then gradually becomes liquid as the temperature approaches the melting point, at which the distance $d$ becomes much smaller than the membrane surface roughness.

Similar measurements were conducted during the melting of microdroplets of an aluminum alloy (AA7075) in contact with the membrane of the calorimetric sensor [25]. Using ultrafast membrane nanocalorimetry combined with high-speed IR thermography, the ITC was measured at heating-cooling rates ranging from $10^3$ K/s to $10^5$ K/s. The measurements were reproducible during many subsequent heating-cooling scans. A gradual increase in the interfacial thermal conductance by an order of magnitude with increasing temperature was found in the range from the solidus temperature $T_{Sol} = 805$ K to the liquidus temperature $T_{Liq} = 901$ K of the alloy. The ITC at the solidus temperature was about $G_{pS} = 3 \cdot 10^4$ W/m$^2$K. Then, as the sample was heated, the ITC gradually increased from $3 \cdot 10^4$ W/m$^2$K at 805 K to $G_{LS} = 1.8 \cdot 10^5$ W/m$^2$K at the liquidus temperature $T_{Liq} = 901$ K [25]. After that, the melting process occurred at $T_{Liq}$ with $G_{LS} = 1.8 \cdot 10^5$ W/m$^2$K. The pre-melting process occurring between $T_{Sol}$ and $T_{Liq}$ was volumetric with enthalpy consumption as shown by direct calorimetric measurements [25].

Near the solidus temperature $T_{Sol} = 805$ K, $G_{pS}(d) = 3 \cdot 10^4$ W/m$^2$K can be represented as the sum of $G_{nf}(d)$ about $10^3$ W/m$^2$K and $G_g(d) = 2.9 \cdot 10^4$ W/m$^2$K at $d$ about 100 nm, $\sigma = 0.6$, $p = 10^5$ Pa, $T_{int} = 820$ K, $\delta = 0.93$ μm, and $\varkappa_g(T_{int}) = 0.057$ W/m · K [58]; see Equations (13) and (15), where $T_{int}$ is the temperature intermediate between $T_S = 805$ K and $T_m = 835$ K. Near the liquidus temperature $T_{Liq} = 901$ K, $G_{LS} = 1.8 \cdot 10^5$ W/m$^2$K can be represented as the sum of $G_{nf}(d) = 1.5 \cdot 10^5$ W/m$^2$K at $d = 6$–20 nm and $G_g(d) = 3 \cdot 10^4$ W/m$^2$K at $d \leq 20$ nm, $\sigma = 0.6$, $p = 10^5$ Pa, $T = 915$ K, $\delta = 1.1$ μm, and $\varkappa_g(T_{int}) = 0.063$ W/m·K [58]; see Equations (13) and (15). Thus, upon the pre-melting of aluminum alloy 7075, the distance $d$ decreases from about 100 nm to 10 nm, while the ITC increases by an order of magnitude. The ITC parameters at the AA7075-membrane interface for the start and end of the pre-melting temperature range are summarized in Table 5.

**Table 5.** ITC parameters at the AA7075-membrane interface at the start and the end of the pre-melting temperature range.

| Sample Temperature $T_S$ K | Intermediate Temperature $T_{int}$ K | ITC $G_{LS}(d)$ W/m$^2$K | ITC Component $G_g(d)$ W/m$^2$K | ITC Component $G_{nf}(d)$ W/m$^2$K | Effective Thickness $d$ nm |
|---|---|---|---|---|---|
| $T_{Sol} = 805$ K | 820 | $3 \cdot 10^4$ | $2.9 \cdot 10^4$ | $10^3$ | 90–160 |
| $T_{Liq} = 901$ K | 433 | $1.8 \cdot 10^5$ | $3 \cdot 10^4$ | $1.5 \cdot 10^5$ | 6–20 |

The gradual change in ITC for AA7075 and pure indium cannot be explained by the gradual change in their viscosity with temperature during the pre-melting process. Indeed, in this case, the magnitude of the ITC change would have to depend on the temperature scan rate $R$. In fact, if the observed change in the ITC were associated with a change in viscosity, then the width of the premelting region should narrow as the temperature scan rate $R$ increases. However, the width of the premelting region does not change with increasing heating rate, at least in the range of $10^3 \text{ K/s} \leq R \leq 10^5 \text{ K/s}$. Moreover, the ITC remains stable after successive heating and cooling cycles. Thus, the gradual change in the ITC is not associated with rheological processes during pre-melting.

However, the physical properties in the bulk of the samples gradually change when heated with the consumption of enthalpy. Indeed, a gradual change in the microstructure upon heating is a natural property of alloys between solidus and liquidus temperatures [69]. Additionally, the pre-melting process with a gradual change in the short-range order was observed when measuring the shear modulus and X-ray diffraction of pure indium and gallium in the temperature range several tens of K below $T_{melt}$ [70–72]. These experiments show that during pre-melting in pure indium and gallium, a significant change in the short-range order and a significant increase in the concentration of vacancies are observed [70–72]. These bulk processes can cause a gradual change in the complex dielectric constant $\varepsilon(\omega)$ of the samples. In turn, the strength of the dispersion forces, the effective distance $d$, and the near-field thermal conductance $G_{nf}(d)$ depend on the dielectric properties of the interacting materials; see Equations (2) and (8). Thus, in the process of pre-melting, there is a significant change in the values of components of the interfacial thermal conductance. This is due to the change in the volumetric microstructure of melting materials with the consumption of enthalpy. A more accurate simulation of the gradual change in the ITC in the pre-melting process is currently not feasible, since it is necessary to obtain the behavior of the complex dielectric functions $\varepsilon(\omega)$ of the measured materials in pre-melting processes. However, our goal is to extend such experiments to various alloys and metals since we believe that our results are universal, at least for alloys in the temperature range between $T_{Sol}$ and $T_{Liq}$, where a gradual change in the microstructure of melting alloys occurs.

## 6. Conclusions

Interfacial thermal conductance is an important parameter for industry, especially for fast thermal processes. During phase transformations of a melting sample, various meso-scopic effects take place at the interface between the sample and the substrate, which affect the interfacial thermal conductance. In this study, we focused on the various contributions to ITC and the changes in these contributions during pre-melting and melting of metal microparticles. The contribution of various mesoscopic effects to the total ITC strongly depends on the effective size of the interfacial gap $d$ between the melting sample and the substrate. During melting, interfacial dispersion forces cause the sample to spread over the surface roughness of the substrate. Thus, the thickness $d$ is significantly reduced. The distance $d$ varies in the range from micrometers to nanometers when the sample is melted. We found that in experiments with metal particles melting on a solid, the main contribution to the ITC is made by the near-field photon tunneling and heat transfer through the gas (the phonon contribution to the ITC is usually not observed). It turns out that the interfacial thermal conductance $G_{LS}$ of the liquid-solid interfaces is mainly due to the near-field thermal conductance $G_{nf}$, while the interfacial thermal conductance $G_{SS}$ of the solid-solid interfaces is due to the thermal conductance through the gas $G_g$. In fact, a sharp jump in ITC by an order of magnitude from $G_{SS}$ to $G_{LS}$ occurs just at the beginning of melting due to the formation of a thin liquid layer on the surface of the sample. However, it was observed that in some metals, the ITC gradually increase upon heating in a certain temperature range below the melting point $T_{melt}$. This unusual gradual change in ITC is observed in a certain temperature range below the melting point during the pre-melting process in samples of aluminum alloy 7075 and pure indium. This pre-melting process occurs in the volume of the samples with enthalpy consumption. Thus, during the pre-melting, the

interfacial thermal conductance gradually changes by an order of magnitude from $G_g(d)$ to $G_{nf}(d)$ as $d$ decreases from micrometer to nanometer scale. We studied the nature of the gradual change in the ITC during the pre-melting process. This gradual change in the ITC during the pre-melting process is associated with a gradual volumetric change in the microstructure of the melting materials. This change in the microstructure during the pre-melting determines the strength of the dispersion forces and, consequently, the effective distance $d$, and the near-field thermal conductance $G_{nf}(d)$. Thus, we have made progress in understanding the effect of pre-melting processes on the ITC of metals and alloys in contact with a solid.

We believe our results are universal, at least for alloys in the temperature range between solidus and liquidus temperatures since the gradual change in microstructure upon heating is a natural property of alloys in this temperature range. It is interesting to extend such experiments to various alloys and pure metals, in which a gradual microstructure change occurs when heated in a certain temperature range below the melting point. Understanding mesoscopic effects and the processes that affect interfacial thermal conductance during the pre-melting and melting of metal microparticles can provide theoretical guidance for optimizing technologies such as laser additive manufacturing.

**Author Contributions:** Conceptualization, A.M.; formal analysis, A.M.; methodology, A.M.; supervision, C.S.; visualization, A.M.; writing—original draft, A.M.; writing—review and editing, C.S. All authors have read and agreed to the published version of the manuscript.

**Funding:** This research received no external funding.

**Institutional Review Board Statement:** Not applicable.

**Informed Consent Statement:** Not applicable.

**Data Availability Statement:** The datasets generated during and/or analyzed during the current study are available from the corresponding author on reasonable request.

**Acknowledgments:** A.M. acknowledges the administrative and technical support of the Prokhorov General Physics Institute of the Russian Academy of Sciences.

**Conflicts of Interest:** The authors declare no conflict of interest.

## Nomenclature

**Latin Symbols**

| | |
|---|---|
| $A$ | Hamaker constant (J) |
| $a$ | interatomic distance (m) |
| $c$ | the speed of light in vacuum (m/s) |
| $c_v$ | heat capacity per one molecule at a constant volume (J/K) |
| $d$ | effective size of the interfacial gap (m) |
| $d_F$ | characteristic length of nonlocal effects in metals (m) |
| $G$ | interfacial thermal conductance (W/m$^2$K) |
| $G_{bb}$ | ITC associated with black-body radiation limit (W/m$^2$K) |
| $G_g(d)$ | ITC due to heat transfer through gas (W/m$^2$K) |
| $G_g^{max}$ | maximum possible ITC due to heat transfer through gas (W/m$^2$K) |
| $G_{nf}(d)$ | ITC due to near-field heat transfer (W/m$^2$K) |
| $G_{nf}^{max}(d)$ | maximum possible ITC due to near-field heat transfer (W/m$^2$K) |
| $G_{ph}(d)$ | ITC due to phonon tunneling (W/m$^2$K) |
| $G_{LS}$ | ITC of liquid-solid interface (W/m$^2$K) |
| $G_{SS}$ | ITC of solid-solid interface (W/m$^2$K) |
| $G_{pS}$ | ITC at pre-melting process (W/m$^2$K) |

| | |
|---|---|
| $Gr$ | Grashof number (dimensionless) |
| $k_B$ | Boltzmann constant (J/K) |
| $k_0$ | wave vector $k_0 = \omega/c$ (m$^{-1}$) |
| $k_{\|}$ | component of the wave vector parallel to the surface (m$^{-1}$) |
| $k_z$ | component of the wave vector perpendicular to the surface (m$^{-1}$) |
| $l_{mfp}$ | mean-free-path of gas molecules (m) |
| $l_e$ | electron mean free path (m) |
| $m$ | mass of a gas molecule (kg) |
| $n$ | number of molecules per unit volume (m$^{-3}$) |
| $Nu$ | Nusselt number (dimensionless) |
| $Pr$ | Prandtl number (dimensionless) |
| $p_c$ | interfacial contact pressure (Pa) |
| $p$ | gas pressure (Pa) |
| $q_g$ | heat flux through the gas (W/m$^2$) |
| $R$ | temperature scan rate (K/s) |
| $R_a$ | arithmetic average of surface roughness (m) |
| $T_{melt}$ | melting point (K) |
| $T_{Liq}$ | liquidus temperature (K) |
| $T_{Sol}$ | solidus temperature (K) |
| $T_S$ | sample temperature (K) |
| $T_m$ | membrane temperature (K) |
| $T_{int}$ | intermediate temperature between $T_S$ and $T_m$ (K) |
| $T_1$ and $T_2$ | temperatures of plate 1 and plate 2 (K) |
| $v_F$ | Fermi velocity (m/s) |
| $V_m$ | average velocity of gas molecules (m/s) |
| $Z$ | number of molecules striking unit area of the surface per unit time (m$^{-2}$s$^{-1}$) |
| **Greek Symbols** | |
| $\gamma$ | specific heats ratio (dimensionless) |
| $\delta$ | distance associated with the surface temperature jump (m) |
| $\varepsilon(\omega)$ | complex dielectric constant (dimensionless) |
| $\varkappa_g$ | gas thermal conductivity (Wm$^{-1}$K$^{-1}$) |
| $\lambda_{mfp}$ | mean free pass of gas molecules (m) |
| $\lambda_0$ | characteristic wavelength of the absorption spectra (m) |
| $\lambda_T$ | characteristic wavelength of thermal radiation (m) |
| $\nu$ | kinematic viscosity of the gas (m$^2$/s) |
| $\rho$ | gas density (kg/m$^3$) |
| $\sigma$ | thermal accommodation coefficient (dimensionless) |
| $\omega$ | angular frequency of electromagnetic waves (rad/s) |
| $\omega_0$ | characteristic frequency of absorption spectra (rad/s) |
| **Special Symbols** | |
| $\hbar$ | reduced Planck constant (J $\cdot$ s) |

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
