# Peer review of "Mesoscopic Effects of Interfacial Thermal Conductance during Fast Pre-Melting and Melting of Metal Microparticles"

_applsci, doi:10.3390/app13127019_

Round 1

Reviewer 1 Report

In the submitted manuscript, the authors are studied the influence of various mesoscopic on the ITC, like the heat transfer through the gas gap, near-field radiative heat transfer and changes in the wetting behavior during melting. Various contributions to the ITC of the liquid-solid interfaces in the processes of fast pre-melting and melting of metal microparticles are studied. The effective distance between materials in contact is a key parameter for determining ITC. This distance change significantly during phase transformations of materials. An unusual gradual change in ITC recently observed during pre-melting below the melting point of some metals is discussed. The pre-melting process does not occur on the surface but is a volumetric change in the microstructure of the materials. This change in the microstructure during the pre-melting determines the magnitude of the dispersion forces, the effective distance, and the near-field thermal conductance. 

The paper is  good structured and includes some new contributions with merits for publication. However, some improvements would still be needed.

-The authors need to give more details of the method that has been used in the paper.

- The results are confused and they need to be more clear, it is better if the authors can summarize the results in tables or as a graphs to be easy to read.

Author Response

Response to the reviewer's comments, applsci-2415835.

The authors are grateful to the reviewers for their valuable comments, which helped us to improve the paper significantly. The corrections have been made, and all changes are marked in the revised manuscript.

In the submitted manuscript, the authors study the influence of various mesoscopic on the ITC, like the heat transfer through the gas gap, near-field radiative heat transfer and changes in the wetting behavior during melting. Various contributions to the ITC of the liquid-solid interfaces in the processes of fast pre-melting and melting of metal microparticles are studied. The effective distance between materials in contact is a key parameter for determining ITC. This distance change significantly during phase transformations of materials. An unusual gradual change in ITC recently observed during pre-melting below the melting point of some metals is discussed. The pre-melting process does not occur on the surface but is a volumetric change in the microstructure of the materials. This change in the microstructure during the pre-melting determines the magnitude of the dispersion forces, the effective distance, and the near-field thermal conductance.

The paper is good structured and includes some new contributions with merits for publication. However, some improvements would still be needed.

-The authors need to give more details of the method that has been used in the paper.

Details are added on pages 11 - 12. The calorimetric sensor consists of an amorphous Si-N membrane (about 1 µm thick) with a resistive heater and a thermocouple sensor located in the central part of the membrane directly below the sample. The temperature sensor, heater and electrical connections are formed by thin-film tracks of doped p-type and n-type polysilicon [25,26]. Ultrafast scanning nanocalorimetry makes it possible to measure the dynamics of the enthalpy absorbed by a sample during fast phase transformations with sub-nanojoule resolution. Measurements can be made at temperature scan rates up to 108 K/s [65]. The dynamics of the temperature on the sample surface opposite the membrane was measured by IR thermography with sub-millisecond time resolution [25,26]. Thus, the temperature difference between the sample and the membrane was measured and used to determine the ITC, since the temperature difference across the sample (less than 1 K) was small compared to the measured temperature difference across the membrane-sample interface. The measurements were completely reproducible during successive heating-cooling temperature scans. The calorimetric measurement of the heat flux through the membrane–sample interface was carried out with an error of about 10%. The ITC measurement error was about 50% or less, depending on the scan rate and temperature range. This error was mainly associated with the uncertainty of the interfacial temperature difference of about 30%. Thus, according to Eq. (1), the ITC was measured during fast phase transformations in metal microdroplets with a sub-millisecond resolution [25,26].

The uncertainty in the distance  estimated from the measured ITC is defined as follows. At distances  100 nm, the ITC is less than , and the ITC is determined mainly by the component , see Fig.1. In this case, the distance  is calculated using equations (13) and (15) with an error of about 45%. This error is mainly due to the uncertainty of the accommodation coefficient , which is about 30%. The error of gas thermal conductivity  (about 3%) is insignificant. This error is associated with the replacement of the thermal conductivity of air with the thermal conductivity of nitrogen gas [58]. The overall error in estimating  is about 70% because ITC is measured with an uncertainty of about 50%. At distances  less than 100 nm, the interfacial thermal conductance  can be represented as the sum of  and , where the near-field component  is estimated as , and  is approximately equal to , see Fig.1. In this case,  is estimated with an uncertainty of about 50%. The uncertainty of  is about 45%, mainly due to the uncertainty of  of about 30%. The total error in estimating  is about 80% because ITC is measured with an uncertainty of about 50%. However, the accuracy of the estimated  is quite satisfactory, since the distance  changes by orders of magnitude in melting experiments (note the logarithmic scales in Fig.1).

- The results are confused, and they need to be more clear, it is better if the authors can summarize the results in tables or as a graphs to be easy to read.

 Improved. The results are summarized in Tables 1 – 5 and Fig.1.

Reviewer 2 Report

In this article, the effects of various mesoscopic effects on Interfacial Thermal Conductance are discussed, the contributions of liquid-solid interfaces to ITC in the processes of fast pre-melting and melting of metal microparticles are investigated,  some theoretical calculations and derivations are carried out. The subject matter of this paper is relatively new, Major revision are recommended, The reasons and suggestions are as follows:

1There are many factors affecting ITC. What is the most important factor?

2There is no chart or table in the whole text, which is unreadable and difficult to understand. It is suggested to add a diagram appropriately.

3The experimental verification content is too little, how reliable is the theoretical calculation? Please give the detailed method and raw data of the experiment.

4Are the results universal when specific samples are used for experimental verification?

5Can relevant simulations be performed to illustrate and verify the results obtained in this paper?

6The Conclusions are too long and the focus of the study is not obvious.

7The overall format and structure of this paper is not like that of a research paper and adjustments are recommended.

average

Author Response

Response to the reviewer's comments, applsci-2415835.

The authors are grateful to the reviewers for their valuable comments, which helped us to improve the paper significantly. The corrections have been made, and all changes are marked in the revised manuscript.

In this article, the effects of various mesoscopic effects on Interfacial Thermal Conductance are discussed, the contributions of liquid-solid interfaces to ITC in the processes of fast pre-melting and melting of metal microparticles are investigated, some theoretical calculations and derivations are carried out. The subject matter of this paper is relatively new, Major revision are recommended, The reasons and suggestions are as follows:

  1. There are many factors affecting ITC. What is the most important factor?

 - Explained on page 2. Interfacial adhesion is the most important factor affecting ITC. For example, interfacial adhesion between nanoparticles and an organic matrix is due to the covalent chemical bonds [18,19]. This chemical adhesion is relatively strong compared to the dispersive adhesion that occurs when metal particles are melted on a solid. Chemical forces act at very short sub-nanometer distances [17]. In this case, acoustic phonons can be the dominant heat carriers at interfaces at sub-nanometer distances [14]. Interfacial phonon tunneling (with an ITC of about 108 W/m2K) was probably observed for aluminum and gold samples specially functionalized with organic liquids [16]. Also, in the case of composites of nanoparticles immersed into an organic matrix, the main contribution to the ITC, about 108 W/m2K, is associated with phonon heat transfer [18,19]. An interfacial thermal resistance (ITR) of about 10–8 m2K/W between nanoparticles and a polymer matrix significantly affects the thermal conductivity of highly thermally conductive CNT/polymer and graphene/polymer composites [18,19]. However, in composites containing nanoparticles with lower thermal conductivity (for example, in composites and nanofluids containing aluminum oxide nanoparticles [20, 21]), an ITR of about 10-8 m2K/W may be insignificant for the overall thermal conductivity of the composite.

In the case of dispersive adhesion, which occurs in experiments on the melting of metal particles on a solid substrate, the ITC is not as high as in the case of chemical adhesion and is usually less than 107 W/m2K. In this study, we focus on the various contributions to ITC and the changes in these contributions during pre-melting and melting of metal microparticles when dispersion adhesion takes place. We found that in experiments with the melting of metal particles on a solid, the main contribution to ITC is made by near-field photon tunneling and heat transfer through the gas (the phonon contribution to the ITC is usually not observed).

  1. There is no chart or table in the whole text, which is unreadable and difficult to understand. It is suggested to add a diagram appropriately.

Improved. Added Tables 1 - 5 and Fig.1.

  1. The experimental verification content is too little, how reliable is the theoretical calculation? Please give the detailed method and raw data of the experiment.

- The experimental results are summarized in Tables 1 – 5 and Figure 1.

- The purpose of this article is to understand the unusual behavior of ITC during the pre-melting process. Raw experimental data are published in ref. [25,26]. However, necessary experimental details and uncertainties are added on pages 11 - 12. The calorimetric sensor consists of an amorphous Si-N membrane (about 1 µm thick) with a resistive heater and a thermocouple sensor located in the central part of the membrane directly below the sample. The temperature sensor, heater and electrical connections are formed by thin-film tracks of doped p-type and n-type polysilicon [25,26]. Ultrafast scanning nanocalorimetry makes it possible to measure the dynamics of the enthalpy absorbed by a sample during fast phase transformations with sub-nanojoule resolution. Measurements can be made at temperature scan rates up to 108 K/s [65]. The dynamics of the temperature on the sample surface opposite the membrane was measured by IR thermography with sub-millisecond time resolution [25,26]. Thus, the temperature difference between the sample and the membrane was measured and used to determine the ITC, since the temperature difference across the sample (less than 1 K) was small compared to the measured temperature difference across the membrane-sample interface. The measurements were completely reproducible during successive heating-cooling temperature scans. The calorimetric measurement of the heat flux through the membrane–sample interface was conducted with an error of about 10%. The ITC measurement error was about 50% or less depending on the scan rate and temperature range. This error was mainly associated with the uncertainty of the interfacial temperature difference about 30%. Thus, according to Eq. (1), the ITC was measured during fast phase transformations in metal microdroplets with a sub-millisecond resolution [25,26].

The uncertainty in the distance  estimated from the measured ITC is defined as follows. At distances  100 nm, the ITC is less than , and the ITC is determined mainly by the component , see Fig.1. In this case, the distance  is calculated using equations (13) and (15) with an error of about 45%. This error is mainly due to the uncertainty of the accommodation coefficient , which is about 30%. The error of gas thermal conductivity  (about 3%) is insignificant. This error is associated with the replacement of the thermal conductivity of air with the thermal conductivity of nitrogen gas [58]. The overall error in estimating  is about 70% because ITC is measured with an uncertainty of about 50%. At distances  less than 100 nm, the interfacial thermal conductance  can be represented as the sum of  and , where the near-field component  is estimated as , and  is approximately equal to , see Fig.1. In this case,  is estimated with an uncertainty of about 50%. The uncertainty of  is about 45%, mainly due to the uncertainty of  about 30%. The total error in estimating  is about 80% because ITC is measured with an uncertainty of about 50%. However, the accuracy of the estimate  is quite satisfactory, since the distance  changes by orders of magnitude in melting experiments (note the logarithmic scales in Fig.1).

  1. Are the results universal when specific samples are used for experimental verification?

Explained in p.15. We believe our results are universal, at least for alloys in the temperature range between solidus and liquidus temperatures, since the gradual change in microstructure upon heating is a natural property of alloys in this temperature range. It is interesting to extend such experiments to various alloys and pure metals, in which a gradual change in microstructure occurs when heated in a certain temperature range below the melting point.

  1. Can relevant simulations be performed to illustrate and verify the results obtained in this paper?

Explained on page 14. A more accurate simulation of the gradual change in the ITC in the pre-melting process is currently not feasible, since it is necessary to obtain the behavior of the complex dielectric functions ε(ω) of the measured materials in pre-melting processes. However, our goal is to extend such experiments to various alloys and metals, since we believe that our results are universal, at least for alloys in the temperature range between  and , where a gradual change in the microstructure of melting alloys occurs.

  1. The Conclusions are too long, and the focus of the study is not obvious.

Corrected on page 14. The conclusions have been revised and shortened.

  1. The overall format and structure of this paper is not like that of a research paper and adjustments are recommended.

Improved. The results are summarized in Tables 1 – 5 and Figure 1.

Reviewer 3 Report

The authors investigated the influence of various mesoscopic effects on the ITC, like the heat transfer through the gas gap, near-field radiative heat transfer and changes in the wetting behavior during melting. Also, various contributions to the ITC of the liquid-solid interfaces in the processes of fast pre-melting and melting of metal microparticles are studied.

The manuscript looks interesting, however, the authors need to carefully address all of the following comments in order to reach the final evaluation:

1. In the context of the presented literature review done by the authors, I really wonder what is the contribution of the current study?! The authors must explain this explicitly in the introduction section in terms of contribution and novelty. 

2. The authors should discuss in the introduction section the importance of their work when considering the following two references: Modeling and sensitivity analysis of thermal conductivity of ethylene glycol-water based nanofluids with alumina nanoparticles, published in Experimental Techniques. And; https://www.sciencedirect.com/science/article/pii/S1359431123000133 

3. It is known that he phonon tunneling provides a significant increase in heat transfer compared to photon tunneling at sub-nanometer gaps. The authors need to explain how their finds support this fact.

4. the authors stated that: It is noteworthy that liquids spread very efficiently over surface roughness even in the absence of good wetting. What is the addition you made to this fact in your study?

Minor

Author Response

Response to the reviewer's comments, applsci-2415835.

The authors are grateful to the reviewers for their valuable comments, which helped us to improve the paper significantly. The corrections have been made, and all changes are marked in the revised manuscript.

The authors investigated the influence of various mesoscopic effects on the ITC, like the heat transfer through the gas gap, near-field radiative heat transfer and changes in the wetting behavior during melting. Also, various contributions to the ITC of the liquid-solid interfaces in the processes of fast pre-melting and melting of metal microparticles are studied.

The manuscript looks interesting; however, the authors need to carefully address all of the following comments in order to reach the final evaluation:

  1. In the context of the presented literature review done by the authors, I really wonder what is the contribution of the current study?! The authors must explain this explicitly in the introduction section in terms of contribution and novelty.
  • Explained on pages 2 - 3. In this study, we focus on the various contributions to ITC and the changes in these contributions during pre-melting and melting of metal microparticles when dispersion adhesion takes place. We found that in experiments with the melting of metal particles on a solid, the main contribution to ITC is made by near-field photon tunneling and heat transfer through the gas (the phonon contribution to the ITC is usually not observed).

It is noteworthy that the melting of a solid is usually initiated from its surface [22], and with the onset of sample melting, a sharp jump in ITC by an order of magnitude occurs [23,24]. However, in alloys [25] and even in pure metals [26], the ITC can gradually increase upon heating in a certain temperature range below the melting point . This recently observed unusual gradual change in ITC is analyzed in this article. It is noteworthy that this pre-melting process does not occur on the surface, but is associated with the absorption of enthalpy in the volume of the sample, as follows from direct calorimetric measurements [25,26]. The purpose of this article is to study the nature of the gradual change in the ITC during the pre-melting process. In this article, it has been established for the first time that the volumetric change in the microstructure of the sample during pre-melting process determines the magnitude of the dispersion forces, the effective distance , and the near-field thermal conductance.

  • Explained in Conclusions on pages 14 - 15. In this study, we focused on the various contributions to ITC and the changes in these contributions during pre-melting and melting of metal microparticles. … We found that in experiments with metal particles melting on a solid, the main contribution to the ITC is made by the near-field photon tunneling and heat transfer through the gas (the phonon contribution to the ITC is usually not observed)…We studied the nature of the gradual change in the ITC during the pre-melting process. This gradual change in the ITC during the pre-melting process is associated with a gradual volumetric change in the microstructure of the melting materials. This change in the microstructure during the pre-melting determines the strength of the dispersion forces and, consequently, the effective distance , and the near-field thermal conductance . Thus, we have made progress in understanding the effect of pre-melting processes on the ITC of metals and alloys in contact with a solid.

We believe our results are universal, at least for alloys in the temperature range between solidus and liquidus temperatures, since the gradual change in microstructure upon heating is a natural property of alloys in this temperature range. It is interesting to extend such experiments to various alloys and pure metals, in which a gradual microstructure change occurs when heated in a certain temperature range below the melting point.

  1. The authors should discuss in the introduction section the importance of their work when considering the following two references: Modeling and sensitivity analysis of thermal conductivity of ethylene glycol-water based nanofluids with alumina nanoparticles, published in Experimental Techniques, and in https://www.sciencedirect.com/science/article/pii/S1359431123000133 .

Discussed on page 2 (references [20,21] are considered). Interfacial adhesion is the most important factor affecting ITC. For example, interfacial adhesion between nanoparticles and an organic matrix is due to the covalent chemical bonds [18,19]. This chemical adhesion is relatively strong compared to the dispersive adhesion that occurs when metal particles are melted on a solid. Chemical forces act at very short sub-nanometer distances [17]. In this case, acoustic phonons can be the dominant heat carriers at interfaces at sub-nanometer distances [14]. Interfacial phonon tunneling (with an ITC of about 108 W/m2K) was probably observed for aluminum and gold samples specially functionalized with organic liquids [16]. Also, in the case of composites of nanoparticles immersed into an organic matrix, the main contribution to the ITC, about 108 W/m2K, is associated with phonon heat transfer [18,19]. An interfacial thermal resistance (ITR) of about 10–8 m2K/W between nanoparticles and a polymer matrix significantly affects the thermal conductivity of highly thermally conductive CNT/polymer and graphene/polymer composites [18,19]. However, in composites containing nanoparticles with lower thermal conductivity (for example, in composites and nanofluids containing aluminum oxide nanoparticles [20, 21]), an ITR of about 10-8 m2K/W may be insignificant for the overall thermal conductivity of the composite. In the case of dispersive adhesion, which occurs in experiments on the melting of metal particles on a solid substrate, the ITC is not as high as in the case of chemical adhesion and is usually less than 107 W/m2K.

  1. It is known that the phonon tunneling provides a significant increase in heat transfer compared to photon tunneling at sub-nanometer gaps. The authors need to explain how their finds support this fact.

- The contribution of phonon tunneling is not achieved in melting experiments, since this contribution can be observed at sub-nanometer distances  when chemical adhesion takes place.

Explained on page 2. In the case of dispersive adhesion, which occurs in experiments on the melting of metal particles on a solid substrate, the ITC is not as high as in the case of chemical adhesion and is usually less than 107 W/m2K. In this study, we focus on the various contributions to ITC and the changes in these contributions during pre-melting and melting of metal microparticles when dispersion adhesion takes place. In experiments with metal particles melting on a solid, we found that the main contributions to ITC are near-field photon tunneling and heat transfer through the gas (the phonon contribution to the ITC is usually not observed).

  1. The authors stated that: It is noteworthy that liquids spread very efficiently over surface roughness even in the absence of good wetting. What is the addition you made to this fact in your study?

- We pay attention to this interesting fact and use it for our analysis.

- Explained on page 10. Thus, in the case of the liquid–solid interfaces, the effective thickness  is much less than the surface roughness , for example, see Table 3. In fact, liquids spread very efficiently over surface roughness even in the absence of good wetting [15,16]. This interesting fact explains the significant difference between the surface roughness  and the effective distance  in experiments on the melting of various materials.

Round 2

Reviewer 2 Report

It's ok

It's ok.